# Development and evaluation of a food frequency questionnaire for use among young children

**Miaobing Zheng**[‡]*, **Karen J. Campbell**[‡], **Emily Scanlan, Sarah A. McNaughton**

School of Exercise and Nutrition Sciences, Institute for Physical Activity and Nutrition (IPAN), Deakin University, Burwood, Victoria, Australia

‡ These authors share first authorship on this work.
* j.zheng@deakin.edu.au

## Abstract

### Background/Objectives

This study described the development of a parent food frequency questionnaire (FFQ) for measuring diets of young children over the past month and the validation of this FFQ against three non-consecutive 24 hour recalls.

### Subjects/Methods

Food and nutrient intakes from a 68-item FFQ were compared with three non-consecutive 24 hour recalls in a follow-up cohort of children aged 1.5, 3.5 and 5.0 years old. Data from both methods were available for 231, 172 and 187 participants at ages 1.5, 3.5 and 5.0 years, respectively.

### Results

Out of 11 nutrients, four (protein, fat, fibre, iron), two (Vitamin C, folate) and three (protein, vitamin C and folate) nutrients showed good-acceptable outcome for 2 out of 3 group-level validation tests at ages 1.5, 3.5 and 5.0 years, respectively. Of 26 food groups, good-acceptable outcome for 2 out of 3 group-level validation tests was revealed for two, four and six food groups at ages 1.5, 3.5 and 5.0 years, respectively. For individual-level validation tests, all nutrients showed good-acceptable outcome for 2 out of 3 individual level tests across three time points, except for folate at age 1.5 years and energy intake at age 3.5 years. Most food groups (22 out of 26) at age 1.5 years and all food groups at both ages 3.5 and 5.0 years showed good-acceptable outcome for 2 out of 3 individual-level validation tests.

### Conclusions

At all three time points, the FFQ demonstrated good-acceptable validity for some nutrients and food groups at group-level, and good-acceptable validity for most nutrients and food

**Data Availability Statement:** Data cannot be shared publicly because of ethical consideration due to potential identifying participant information like local government area code and postcode. The

Deakin University Human Ethics Committee imposed that the data can be accessed for research reasons only and whoever uses the data needs to get ethical approval or apply a waiver for secondary data analysis. Data are available from the Deakin University Institutional Data Access (contact via gavin.abbott@deakin.edu.au) for researchers who meet the criteria for access to the confidential data.

**Funding:** MZ is supported by an Australian National Health Research Medical Council Early Career Fellowship (GNT1124283).

**Competing interests:** The authors have declared that no competing interests exist.

groups at individual-level. This quantitative FFQ is a valid and robust tool for assessing total diet of young children and ranking individuals according to nutrient and food intakes.

## Introduction

The prevention of obesity in childhood has become an international health priority [1]. Obesity is recognised to have numerous negative impacts on children's health and wellness during childhood and adult life [2]. Diet plays an important role in the aetiology and prevention of obesity and dietary behaviours in childhood track through the lifespan and have impacts on adult health [3–7]. However, the evaluation of early dietary behaviours in obesity prevention is limited [8] by the lack of suitable dietary assessment tools among toddlers and young children [9–12]. Well-designed methods suitable for determining dietary intake in toddlers and young children are required to adequately evaluate obesity prevention efforts and provide the evidence base for nutrition policies and programs [13].

Widely used dietary assessment methods include 24-hour recalls, food records or diaries and food frequency questionnaires (FFQs) [11]. Methods such as 24-hour recalls and food records are resource intensive and are associated with high subject burden [14]. FFQs are advantageous as they involve low respondent burden, can be self-administered, and require less training of research staff [15]. FFQs can provide useful measures of dietary intake suitable for large scale epidemiological studies and investigating diet and diseases relationships [16].

A major consideration is the need for FFQs to be suitable for the target population [15] and tailored to the population under study with respect to culture, age, sex and other determinants of food intake. Several FFQs have been developed for Australian young children to assess intakes of specific food groups or nutrients rather than total diet including both absolute food and nutrient intakes [17–21]. Flood et al., [17] and Bennett et al., [18] developed and evaluated short FFQs for measuring daily serving of food and beverage intakes. Devenish et al., [21] validated a tool to assess total and free sugars intakes of Australian toddlers. Only one quantitative FFQ exists [20], but the tool was not developed specifically for young children, rather it was modified from a quantitative FFQ for older children and adolescents. Therefore, a quantitative FFQ developed specifically for Australian young children is needed. The present study aimed to describe the development and evaluation of a quantitative FFQ to assess the food and nutrient intakes in young children aged 1.5 to 5 years in Australia using 3 x 24-hour recalls among children participating in Melbourne InFANT study.

## Materials/Subjects and methods

### Participants

This study is nested in the Melbourne Infant Feeding Activity and Nutrition Trial (InFANT) Program, a 15-month cluster-randomised controlled trial involving first-time parents for childhood obesity prevention, with additional follow-up until 5 years old with no intervention [22–24]. First-time parents' groups (n = 62) were recruited in 2008 from across Melbourne via Maternal and Child Health Nurses, as has been described elsewhere, with a total of 542 parent-child pairs participating at baseline [22]. To evaluate the nutrition components, child dietary intakes were assessed at child ages 9 months, 1.5, 3.5 and 5 years of age, commencing in 2008 and concluding in 2013, via three non-consecutive 24-hour recalls with parents [25]. For the purpose of this validation study, main carers completed the self-administered FFQ when children were 1.5, 3.5 and 5 years. Renewed written parental consent was provided at each time

point. Ethics approval was granted by the Deakin University Ethics Committee (ID number: EC 175–2007) and by the Victorian Office for Children (Ref: CDF/07/1138).

## 24-hour recalls

At each time point, child diets were assessed via three telephone-administered 24-hour recalls with parents by trained dietitians [26]. Recalls were conducted on non-consecutive days including two weekdays and one weekend day using a 5-pass standardized recall process based on the method validated by the United States Department of Agriculture [27] and the method used by the United States' Feeding Infants and Toddlers Study [28]. During the interview, parents were asked to recall all food and beverage consumed by the child over the previous day (24-hours). Purpose-designed food measurement booklets were provided to assist parents with estimation of quantities consumed. Recalls were unscheduled where possible (96% of recalls). Given that many children spent time with carers other than their main carer by 1.5 years of age, it was sometimes necessary to involve other carers in the reporting of children's diets. For children who spent less than two days per week with another carer, diet recalls were conducted only on days *after* the main carer had been with their child. For children who spent more than 3 days with another carer, up to 2 recalls was pre-scheduled (i.e. parents were told the date that we would conduct the recall). When recalls were pre-scheduled, the alternative carer was asked to record the child's food and beverage intake while the child was in care using a purpose-designed food diary and the main carer used the diary to report the child's intake during the period of the day they were in care. Data was coded by trained researchers using an in-house, purposed designed database incorporating the Australian Food Supplement and Nutrient Database ("AUSNUT2007") [29], with additional infant-specific products added to derive food and nutrient intakes. Coding of all recalls was checked for accuracy and completeness by a dietitian. The average nutrient and food intakes from the three 24 hour recall at each time point was calculated for comparison with the FFQ.

## Food frequency questionnaire (FFQ)

The FFQ food list was developed using data on 2–5 year olds from the 2007 National Children's Nutrition and Physical Activity Survey (NCNPAS), which was the most recent and comprehensive data on dietary intakes of Australian children available at the time. Initially, data analysis involved combining all the individual foods reported on the questionnaire according to nutrient profile and culinary use into food items and groups suitable for inclusion in the questionnaire. Stepwise regression techniques were then applied to identify those foods which explained the most variation in intakes of key nutrients, targeting at least 80% of variation in nutrient intakes [30, 31]. Key nutrients to be assessed were those relevant to obesity prevention (energy, fat, saturated fat, sugars) and key indicators of diet quality among young children (protein, fibre, folate and vitamin C as markers of fruit and vegetable intake, and iron and calcium).

The questionnaire was comprised of two sections similar in format to previously developed FFQs commonly used in Australia [32]. The first section contained questions relating to general eating habits including usual intake of fruit and vegetables, type of milk and bread usually consumed and supplement use. The second section assessed the frequency of consumption of 68 food items over the past month with nine response options ('never or less than once a month' to '6 or more times a day'). A copy of the FFQ is provided in the S1 File.

Food and nutrients intakes from the FFQ were calculated using a purposed designed database incorporating AUSNUT2007 food composition database. The database was developed by matching each FFQ item to one or more foods from AUSNUT2007, in order to generate a

nutrient profile for each FFQ item. Portion sizes for each FFQ item were based on median portion sizes consumed at each age in the 2007 NCNPAS data and applied to each FFQ item. The resulting database was utilised to convert the data collected on frequency of consumption from the questionnaire into the amount of each food item consumed in grams and to calculate the nutrient intake for each participant. Daily consumption of each FFQ food item in grams was calculated by converting the frequency of consumption into daily equivalents (Never = 0; 1-3/month = 0.067; 1/week = 0.143; 2-4/week = 0.429; 5-6/week = 0.786; 1/day = 1.0; 2-3/day = 2.5; 4-5/day = 4.5; $\geq$6/day = 6.0) and then multiplying by the calculated median portion size for that food.

## Covariates

A parent questionnaire was conducted at each time point to collect information on child age, sex and maternal education. Maternal education was categorised as low (completed up to year 12), intermediate (completed trade/certificate post-secondary school) and high (completed university degree or beyond). Children's anthropometrics were measured in light clothing by trained staff. Weight was measured to 10 grams using calibrated infant digital scales (Tanita 1582- Tokyo, Japan). Height/length was measured to 0.1cm using calibrated measuring mat (Seca 210, Seca Deutschland, Germany) or portable stadiometer (Invicta IPO955, Oadby, Leicester). Body mass index (BMI) z-scores were calculated using WHO growth standards [33].

## Analytical sample

Participants were included in this analysis if they had three 24-hour recalls, complete FFQ data (<10% of items from the FFQ missing), complete data for covariates used to describe the sample (child age, sex, and BMI z-score and maternal education), had not consumed breastmilk or had energy intakes within ±3SD from the mean after exclusion criteria was applied, the final sample for the validation study was 231 at age 1.5 years, 172 at age 3.5 years and 187 at age 5 years (Fig 1). Comparison of children who included versus excluded from the analysis is presented in S1 Table. Children who included in the analysis were slightly younger than those excluded from the analysis with mean difference ranging from 0.1 to 0.3 years. However, no significant difference was found in regards to sex, maternal education, and BMI z-score.

## Statistical analysis

At each time point, nutrient and food groups intakes from the FFQ were compared against the average of the 24-hour recalls at each age. To enable comparison between methods, FFQ food items classified into 26 food groups and foods from the 24-hour recall data were matched to the most appropriate food group. Detailed categorisation of 26 food groups is provided in S2 Table.

 Group-level and individual-level validation tests with implications for research and clinical setting, respectively, were conducted to test differences between the FFQ and 24-hour recall measures [34, 35]. Three group-level validation tests were performed: paired t-test/Wilcoxon signed rank sum test, Bland-Altman correlation between mean and mean difference, and Bland-Altman limit of agreement (LOA) index. Mean and standard errors were calculated for the nutrient intakes estimated from each method and compared using paired t-test; skewed data were log-transformed. Median and interquartile range were calculated for food group intakes, Wilcoxon signed rank sum test was used to test differences between the measures. Bland-Altman correlation between mean and mean difference of two measures examines the presence of proportional bias as well as agreement at the group level with a significant P-value

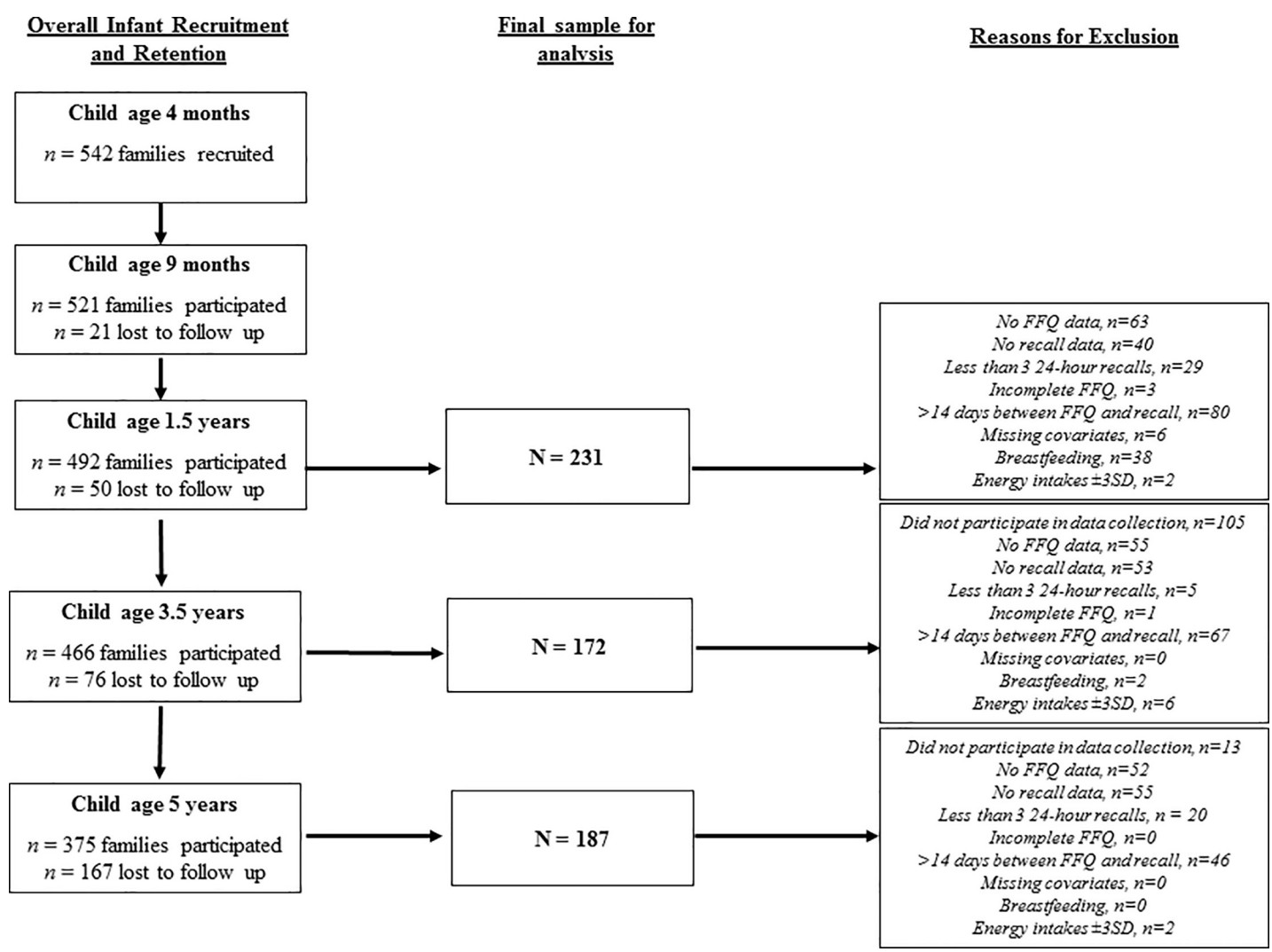

**Fig 1. Flowchart of recruitment and exclusion of participants in the Melbourne InFANT program for FFQ validation and analyses.**

≤0.05 indicating the presence of bias. Bland-Altman LOA index was calculated as the percentage of individuals with values outside of the limit of agreement (<5% being acceptable) [34]. The individual-level agreement was assessed by three validation tests: correlation coefficient, categorical agreement and weighted kappa. Pearson's and Spearman's correlation (r) analyses were performed for nutrient and food group intakes respectively to assess strength and direction of association at individual level. Strength of correlation was defined as poor (<0.20), acceptable (0.20–0.49), and good (≥0.50) outcomes [34]. To account for random within person variation among three 24-hour recalls, the energy-adjusted correlation coefficients were de-attenuated based on the formula: $R = r\sqrt{1 + \lambda_x/n_x}$, in which the attenuation factor ($\lambda_x$) is the ratio of within- and between-person variances obtained from the three recalls and $n_x$ equals to 3 [31]. Categorical agreement including chance at individual level was assessed by grouping participants into quintiles to identify the proportion of participations correctly classified within one quintile (≥50% good outcome)[34]. Categorical agreement excluding chance at individual level was further assessed using weighted kappa statistics. Degrees of agreement for the weighted kappa values ($K_w$) were characterised as poor agreement (Kw<0.20),

acceptable agreement (Kw = 0.20–0.59), and good agreement (Kw≥0.60)[34, 35]. Nutrient and food intakes from both FFQ and 24-hour recall were energy-adjusted to remove correlated measurement error using the residual method with nutrient and food intakes regressed on total energy intake [36]. Equality of average correlation coefficients between each time point was tested by "CORTESTI" command. All statistical analyses were performed using STATA 12.1 (StataCorp, USA). Significance level was set at p<0.05.

## Results

### Sample characteristics

At ages 1.5, 3.5 and 5.0 years, the percentage of males were 55.4%, 48.8% and 50.8% and the mean BMI z-score was 0.84, 0.70 and 0.61, respectively (Table 1). Consistent with the larger InFANT sample, mothers in the validation study were highly educated with more than half the sample completing university degrees or higher (55.4–62.0%). The mean number of days between the two dietary assessment measures were 4.0, 6.7 and 6.5 days at ages 1.5, 3.5 and 5.0 years, respectively. The assessment period between three 24-hour recalls and FFQ is also showed in S1 Fig.

### Nutrient intakes

Comparison of energy-adjusted nutrient intakes estimated from FFQ and 24-hour recalls, and group-level validation analyses results are showed in Table 2. At age 1.5 years, mean FFQ nutrient intake estimates were significantly higher than recall estimates for all nutrients excluding folate (Table 2). At ages 3.5 and 5.0 years, the FFQ overestimated most nutrient intakes with the exception of fat, saturated fat and folate when compared with the recall estimates. FFQ estimates of fat and saturated fat were slightly smaller than the recall estimates, whereas, FFQ and recall estimates of folate were similar (P>0.05) at ages 3.5 and 5.0 years. For Bland-Altman correlation of mean and mean difference, most nutrients at ages 1.5 (8 out of 11) at 5 (6 out of 11) years exhibited no proportional bias (P>0.05). In contrast, at age 3.5 years, 9 out of 11 nutrients revealed bias. Of 11 nutrients assessed, 6, 5 and 4 nutrients showed acceptable outcome for Bland-Altman LOA index at age 1.5, 3.5 and 5 years, respectively. Four (protein, fat, fibre, iron), two (Vitamin C, folate) and three (protein, vitamin C and folate) nutrients at ages 1.5, 3.5 and 5.0 years, respectively, showed good-acceptable outcome for 2 out of 3 group-level validation tests.

**Table 1. Characteristics of participants at each time-point.**

| Variable | Age 1.5 years | Age 3.5 years | Age 5.0 years |
|---|---|---|---|
| Sample size, n | 231 | 172 | 187 |
| Sex, n (%) male | 128 (55.4%) | 84 (48.8%) | 95 (50.8%) |
| Maternal education, n (%)* | | | |
| Low | 51(22.1%) | 30 (17.4%) | 34 (18.2%) |
| Intermediate | 52 (22.5%) | 36 (20.9%) | 37 (19.8%) |
| High | 128 (55.4%) | 106 (61.6%) | 116 (62.0%) |
| Age of child | 1.5 (0.01) | 3.6 (0.01) | 5.0 (0.01) |
| Body Mass Index z-score, mean (SE) | 0.84 (0.06) | 0.70 (0.06) | 0.61 (0.07) |
| Number of days between FFQ and 24-hour recalls, mean (SE) | 4.0 (0.2) | 6.7 (0.3) | 6.5 (0.3) |

*Low–completed up to year 12, Intermediate–completed trade/certificate post-secondary school, High–completed university degree or beyond, FFQ: quantitative food frequency questionnaire

**Table 2. Group-level validation tests comparing energy-adjusted nutrient intakes estimated by the food frequency questionnaire (FFQ) and the 24-hour recall at each time-point†.**

| Nutrient | Age 1.5 years(n = 231) | | | | Age 3.5 years(n = 172) | | | | Age 5.0 years(n = 187) | | | |
|---|---|---|---|---|---|---|---|---|---|---|---|---|
| | FFQ | 24-Hour recall | Bland-Altman Pearson *r* P-value | Bland-Altman's LOA index | FFQ | 24-Hour recall | Bland-Altman Pearson *r* P-value | Bland-Altman's LOA index | FFQ | 24-Hour recall | Bland-Altman Pearson *r* P-value | Bland-Altman's LOA index |
| Energy (kJ) | **5047 (79)** | **4410 (55)** | <0.001 | 6.9% | **5787 (113)** | **5318 (84)** | 0.002 | 4.7% | **6444 (132)** | **5883 (87)** | <0.001 | 7.0% |
| Protein (g) | **57.1 (0.4)** | **45.8 (0.4)** | 0.23 | 3.9% | **63.9 (0.5)** | **54.4 (0.6)** | 0.001 | 5.2% | **68.0 (0.6)** | **59.7 (0.7)** | 0.08 | 4.8% |
| Carbohydrate (g) | **147.1 (0.9)** | **129.0 (0.9)** | 0.76 | 6.5% | **176.7 (1.2)** | **156.6 (1.6)** | 0.001 | 5.8% | **203.1 (1.5)** | **174.9 (1.4)** | 0.43 | 6.4% |
| Sugars (g) | **71.6 (0.9)** | **68.1 (0.9)** | 0.87 | 7.4% | **83.3 (1.3)** | **78.3 (1.3)** | 0.99 | 6.4% | **98.2 (1.7)** | **83.4 (1.4)** | 0.01 | 5.3% |
| Fat (g) | **39.9 (0.4)** | **37.2 (0.4)** | 0.65 | 3.9% | **42.5 (0.5)** | **44.6 (0.6)** | 0.004 | 4.7% | **45.8 (0.6)** | **48.7 (0.6)** | 0.49 | 5.9% |
| Saturated fat (g) | **19.8 (0.3)** | **18.8 (0.3)** | 0.49 | 4.3% | **19.4 (0.3)** | **20.3 (0.4)** | 0.02 | 5.8% | **20.5 (0.4)** | **21.4 (0.3)** | 0.04 | 5.9% |
| Fibre (g) | **16.2 (0.2)** | **13.0 (0.2)** | 0.85 | 4.8% | **20.2 (0.3)** | **16.1 (0.3)** | 0.047 | 7.0% | **23.3 (0.4)** | **18.3 (0.3)** | 0.16 | 5.3% |
| Iron (mg) | 7.1 (0.1) | **6.6 (0.1)** | 0.34 | 3.0% | **8.1 (0.1)** | 7.2 (0.2) | 0.003 | 4.1% | **8.9 (0.1)** | **8.1 (0.2)** | <0.001 | 3.7% |
| Calcium (mg) | 862.1 (10.5) | 758.9 (11.1) | 0.40 | 5.2% | 814.3 (12.4) | 729.1 (15.7) | 0.002 | 6.4% | 803.4 (14.1) | 766.2 (14.6) | 0.65 | 5.3% |
| Vitamin C (mg)‡ | **74.0 (69.2–79.2)** | **44.2 (40.8–48.0)** | <0.001 | 6.1% | **95.1 (86.5–104.6)** | **60.5 (55.3–66.1)** | 0.24 | 4.7% | **112.2 (103.0–122.2)** | **65.0 (59.8–70.6)** | 0.77 | 3.7% |
| Folate (ug) | **227.1 (3.1)** | **266.5 (6.3)** | <0.001 | 4.8% | 285.6 (277.6–293.8) | 288.8 (273.7–304.8) | <0.001 | 2.9% | 311.2 (302.4–320.3) | 326.7 (311.2–343.0) | <0.001 | 3.7% |

†Values were presented as either mean (standard error) or median (interquartile range), Bland-Altman Pearson's correlation *r* P-value (P≤0.05 presence of bias), Bland-Altman Limits of Agreement (LOA) index (<5% being acceptable); Bold indicates two measures estimated by FFQ and 24hr recall were significant different using paired t-test or Wilcoxon signed rank sum test, adjusted for energy intake

‡Log-transformed, values in parentheses are 95% CI

Table 3 illustrates the individual-level validation test results for FFQ and 24-hour recall energy-adjusted nutrient intakes. Out of 11 nutrients assessed, most nutrients exhibited acceptable correlation (r>0.20) at ages 1.5 (n = 10), 3.5 (n = 9) and 5.0 (n = 10) years. The mean Pearson's r for nutrient intakes was 0.30, 0.33 and 0.32 for ages 1.5, 3.5 and 5.0 years, respectively. Comparison of equality of average Pearson's r between time points revealed no significant difference. Accounting for within person variation, the Pearson correlation at all time points for all nutrients improved, and the average de-attenuated Pearson's R was 0.34, 0.38 and 0.37 at ages 1.5, 3.5 and 5 years, respectively. Percentage agreement of nutrient intakes between the FFQ and recall (agreement within 1 quintile) showed good agreement (>50%) with average percentage agreement of 62.7%, 64.7%, and 62.4%, at age 1.5, 3.5, 5.0 years, respectively. Weighted kappa values showed agreement ranging from poor to acceptable with Kw of 0.06 to 0.34 across three time point. The mean Kw at age 1.5, 3.5 and 5.0 years was 0.19, 0.21, and 0.19, respectively. Ten out of 11 nutrients at ages 1.5 and 3.5 years, and all nutrients at age 5.0 years exhibited good-acceptable agreement for 2 out of 3 individual-level validation tests. The number of nutrients showed good-acceptable agreement for all three individual level tests was 3, 7, and 5 at ages 1.5, 3.5 and 5.0 years, respectively.

**Table 3. Individual-level validation tests comparing energy-adjusted nutrient intakes estimated by the food frequency questionnaire and the 24-hour recall at each time-point†.**

| Nutrient | Age 1.5 years(n = 231) | | | | Age 3.5 years (n = 172) | | | | Age 5.0 yearsT3(n = 187) | | | |
|---|---|---|---|---|---|---|---|---|---|---|---|---|
| | r | R | % agreement | $K_w$ | r | R | % agreement† | $K_w$ | r | R | % agreement | $K_w$ |
| Energy (kJ) | 0.22 | 0.25 | 62.8 | 0.19 | 0.14 | 0.16 | 57.6 | 0.12 | 0.20 | 0.23 | 59.9 | 0.12 |
| Protein (g) | 0.29 | 0.34 | 60.6 | 0.20 | 0.50 | 0.58 | 72.1 | 0.34 | 0.52 | 0.60 | 66.8 | 0.31 |
| Carbohydrate (g) | 0.26 | 0.30 | 64.5 | 0.19 | 0.43 | 0.49 | 68.6 | 0.27 | 0.26 | 0.29 | 64.2 | 0.17 |
| Sugars (g) | 0.27 | 0.30 | 61.0 | 0.18 | 0.23 | 0.26 | 56.4 | 0.06 | 0.27 | 0.31 | 58.3 | 0.15 |
| Fat (g) | 0.27 | 0.31 | 60.2 | 0.16 | 0.39 | 0.44 | 65.1 | 0.22 | 0.23 | 0.26 | 62.0 | 0.17 |
| Saturated fat (g) | 0.41 | 0.46 | 61.5 | 0.18 | 0.40 | 0.45 | 64.5 | 0.23 | 0.43 | 0.49 | 63.1 | 0.24 |
| Fibre (g) | 0.44 | 0.49 | 68.8 | 0.27 | 0.47 | 0.53 | 66.9 | 0.26 | 0.41 | 0.46 | 63.6 | 0.22 |
| Iron (mg) | 0.47 | 0.52 | 68.8 | 0.28 | 0.46 | 0.51 | 71.5 | 0.33 | 0.31 | 0.34 | 62.6 | 0.22 |
| Calcium (mg) | 0.34 | 0.38 | 64.5 | 0.19 | 0.24 | 0.27 | 62.2 | 0.16 | 0.41 | 0.46 | 64.2 | 0.22 |
| Vitamin C (mg) | 0.26 | 0.30 | 64.1 | 0.19 | 0.22 | 0.25 | 68.0 | 0.27 | 0.33 | 0.38 | 63.1 | 0.18 |
| Folate (ug) | 0.03 | 0.04 | 53.3 | 0.06 | 0.18 | 0.20 | 59.3 | 0.10 | 0.18 | 0.21 | 58.8 | 0.12 |

†Pearson's correlation r, De-attenuated Pearson's correlation R (<0.2 poor, 0.2–0.49 acceptable, ≥0.5 good), Percentage agreement within 1 quintile (≥50% good), $K_w$: weighted kappa (<0.20 Poor, 0.20–0.59 acceptable, ≥0.60 good)

## Food intakes

Comparison of energy-adjusted median food intakes (g/day) estimated by FFQ and 24-hour recall by Wilcoxon signed rank sum test are shown in S3 Table. Of 26 food groups examined, percentage (number) of food groups at ages 1.5, 3.5 and 5.0 years that had median FFQ estimates significantly higher than recall estimates was 77% (20), 65% (17) and 62% (16), respectively. Similar FFQ and recall intake estimates were found for four food categories, namely milk, cream/ice-cream/custard, white bread, and sugar/jams/honey at age 1.5 years. At age 3.5 years, eight food categories including cheese, white bread, breakfast cereal, eggs, potato, hot chips, takeaway style foods, and sweet snack foods had similar intake estimates between two methods (P>0.05). At age 5.0 years, two methods produced similar intake estimates for seven food categories: all other beverage, yoghurt, cream/ice-cream/custard, white bread, poultry, potato and savoury snack foods. Results of two other group-level validation tests are reported in Table 4. The number of food groups showed no proportional bias (Bland-Altman correlation P≤0.05) was fairly consistent across time point (9, 10 and 9 out of 26 respectively for ages 1.5, 3.5 and 5.0 years). For Bland-Altman LOA index, 7, 9, 12 food groups at ages 1.5, 3.5 and 5.0 years, respectively showed acceptable outcome (< 5%). Two (water, sugars/jams/honey), four (eggs, potato, hot chips, crispbread/crackers) and six (all other beverage, yoghurt, cream/ice cream/custard, white bread, poultry, takeaway style foods) food groups at ages 1.5, 3.5 and 5.0 years, respectively, showed good-acceptable outcome for 2 of 3 group-level validation tests

Individual-level validation analyses results for energy-adjusted food intakes estimated by FFQ and 24-hour recall are presented in Table 4. Mean Spearman's r among all food groups was 0.31, 0.34, and 0.34 respectively at ages 1.5, 3.5 and 5.0 years. All food groups at age 3.5 years and most food groups at ages 1.5 (21/26) and 5.0 (24/26) years revealed acceptable correlations (r≥0.20). No significant difference between Spearman's r between time points were found. Spearman correlation for all food intakes at all time points improved after correcting for within person variation with average de-attenuated Spearman's R of 0.36, 0.40, and 0.39 at ages 1.5, 3.5, and 5.0 years, respectively. The percentage agreement of all food intakes (agreement within 1 quintile) across all time points was above 50% with average percentage agreement of 62.2%, 64.0% and 64.2% at age 1.5, 3.5 and 5.0 respectively (Table 5). Weighted kappa for food

**Table 4. Group-level validation tests comparing energy-adjusted food intakes estimated by the food frequency questionnaire and the 24-hour recall at each time-point†.**

| Food items | Age 1.5 years (n = 231) | | Age 3.5 years (n = 172) | | Age 5.0 yearsT3(n = 187) | |
|---|---|---|---|---|---|---|
| | Bland-Altman Spearman $r_s$ P-value | Bland-Altman LOA index | Bland-Altman Spearman $r_s$ P-value | Bland-Altman LOA index | Bland-Altman Spearman $r_s$ P-value | Bland-Altman LOA index |
| Water | 0.20 | 4.8% | 0.04 | 3.5% | 0.02 | 2.1% |
| Milk | <0.001 | 5.2% | 0.03 | 5.8% | 0.002 | 3.7% |
| All other beverages | 0.01 | 3.0% | <0.001 | 7.0% | 0.01 | 4.3% |
| Cheese | 0.52 | 6.0% | <0.001 | 6.4% | <0.001 | 4.8% |
| Yoghurt | 0.34 | 6.1% | 0.31 | 6.4% | 0.001 | 4.3% |
| Cream/ ice-cream/ custards | 0.01 | 6.1% | 0.55 | 2.9% | 0.06 | 4.3% |
| Non-white bread | 0.44 | 7.8% | <0.001 | 6.4% | <0.001 | 5.3% |
| White bread | 0.09 | 6.9% | 0.02 | 7.6% | 0.75 | 7.5% |
| Breakfast cereal | <0.001 | 6.0% | <0.001 | 7.6% | <0.001 | 5.9% |
| Rice/pasta/other cereals | 0.001 | 3.9% | 0.001 | 4.7% | 0.004 | 5.3% |
| Red meat | 0.53 | 5.6% | 0.90 | 5.8% | 0.65 | 5.3% |
| Poultry | 0.04 | 6.9% | 0.25 | 5.8% | <0.001 | 4.8% |
| Seafood | 0.11 | 6.5% | 0.047 | 6.4% | 0.06 | 6.4% |
| Processed meat | <0.001 | 5.2% | <0.001 | 5.2% | 0.015 | 5.3% |
| Eggs | 0.02 | 6.5% | 0.003 | 1.1% | 0.004 | 6.4% |
| Fruit | 0.002 | 5.2% | 0.09 | 7.6% | <0.001 | 4.3% |
| Vegetables (no potatoes) | 0.19 | 6.5% | 0.24 | 5.8% | 0.103 | 7.0% |
| Potato | 0.01 | 5.2% | 0.02 | 4.7% | 0.001 | 5.3% |
| Hot chips | 0.004 | 5.2% | 0.09 | 4.7% | 0.27 | 9.1% |
| Takeaway style foods | 0.61 | 5.2% | 0.02 | 8.1% | 0.08 | 4.3% |
| Sweet snack foods | <0.001 | 4.3% | <0.001 | 5.2% | <0.001 | 4.3% |
| Savoury snack foods | <0.001 | 5.2% | 0.09 | 4.7% | <0.001 | 4.3% |
| Crispbreads and crackers | <0.001 | 6.1% | 0.65 | 4.1% | 0.003 | 7.0% |
| Nuts and seeds | <0.001 | 3.9% | 0.58 | 5.2% | 0.45 | 7.5% |
| Butter and margarine | <0.001 | 4.8% | 0.01 | 5.2% | 0.052 | 5.9% |
| Sugars, jams and honey | 0.01 | 4.3% | <0.001 | 4.7% | <0.001 | 4.3% |

†Bland-Altman Spearman's correlation r P-value ≤0.05 (presence of proportional bias), Bland-Altman Limits of Agreement (LOA) index (<5% being acceptable)

intakes showed poor to acceptable agreement ranging from 0.05 to 0.47 across three time point. The mean Kw value at age 1.5, 3.5 and 5.0 years was 0.21, 0.22, and 0.22 respectively. Most food groups (22 out of 26) at age 1.5 years and all food groups at both ages 3.5 and 5.0 years showed good-acceptable outcome for 2 out of 3 individual-level validation tests. Similar number of food groups (15 at age 1.5 years, 14 at age 3.5 years and 13 at age 5.0 years) revealed good-acceptable outcome for all three individual-level validation tests across time points.

## Discussion

The present study described the evaluation of a parent FFQ to assess the nutrient and food group intakes in young children aged 1.5 to 5 years in Australia using 3 x 24-hour recalls. At

**Table 5. Individual-level validation tests comparing energy-adjusted food intakes estimated by the food frequency questionnaire and the 24-hour recall at each time-point†.**

| Food item | Age 1.5 years (n = 231) | | | | Age 3.5 years (n = 172) | | | | Age 5.0 years (n = 187) | | | |
|---|---|---|---|---|---|---|---|---|---|---|---|---|
| | $r_s$ | $R_s$ | % agreement | $K_w$ | $r_s$ | $R_s$ | % agreement | $K_w$ | $r_s$ | $R_s$ | % agreement | $K_w$ |
| Water | 0.38 | 0.40 | 69.7 | 0.28 | 0.42 | 0.46 | 64.0 | 0.23 | 0.44 | 0.47 | 68.5 | 0.31 |
| Milk | 0.30 | 0.33 | 62.3 | 0.18 | 0.47 | 0.51 | 67.4 | 0.27 | 0.52 | 0.57 | 73.3 | 0.34 |
| All other beverages | 0.27 | 0.30 | 51.5 | 0.14 | 0.53 | 0.58 | 75.6 | 0.34 | 0.65 | 0.71 | 81.3 | 0.47 |
| Cheese | 0.49 | 0.56 | 69.7 | 0.32 | 0.29 | 0.33 | 57.6 | 0.13 | 0.36 | 0.43 | 64.2 | 0.20 |
| Yoghurt | 0.53 | 0.60 | 70.6 | 0.36 | 0.53 | 0.61 | 73.3 | 0.34 | 0.54 | 0.61 | 73.8 | 0.33 |
| Cream/ ice-cream /custards | 0.44 | 0.51 | 67.1 | 0.32 | 0.25 | 0.29 | 55.8 | 0.16 | 0.26 | 0.31 | 59.4 | 0.19 |
| Non-white bread | 0.25 | 0.28 | 62.3 | 0.17 | 0.32 | 0.37 | 63.4 | 0.19 | 0.41 | 0.46 | 65.2 | 0.24 |
| White bread | 0.31 | 0.35 | 64.5 | 0.22 | 0.22 | 0.25 | 52.9 | 0.09 | 0.34 | 0.39 | 65.8 | 0.25 |
| Breakfast cereal | 0.19 | 0.21 | 60.6 | 0.15 | 0.29 | 0.33 | 62.8 | 0.18 | 0.22 | 0.25 | 63.1 | 0.19 |
| Rice/pasta /other cereals | 0.35 | 0.41 | 67.5 | 0.20 | 0.33 | 0.39 | 64.5 | 0.19 | 0.26 | 0.30 | 58.3 | 0.13 |
| Red meat | 0.12 | 0.15 | 56.3 | 0.06 | 0.28 | 0.34 | 57.0 | 0.15 | 0.18 | 0.21 | 63.1 | 0.14 |
| Poultry | 0.15 | 0.18 | 59.3 | 0.12 | 0.25 | 0.31 | 64.5 | 0.18 | 0.25 | 0.29 | 60.0 | 0.15 |
| Seafood | 0.33 | 0.33 | 63.2 | 0.25 | 0.31 | 0.38 | 62.2 | 0.22 | 0.33 | 0.39 | 63.1 | 0.20 |
| Processed meat | 0.33 | 0.38 | 67.1 | 0.26 | 0.35 | 0.42 | 68.0 | 0.24 | 0.36 | 0.43 | 63.1 | 0.18 |
| Eggs | 0.34 | 0.41 | 60.2 | 0.21 | 0.33 | 0.39 | 64.0 | 0.22 | 0.32 | 0.38 | 63.6 | 0.19 |
| Fruit | 0.33 | 0.37 | 65.8 | 0.24 | 0.41 | 0.46 | 67.4 | 0.30 | 0.43 | 0.49 | 67.9 | 0.32 |
| Vegetables (no potatoes) | 0.23 | 0.27 | 59.3 | 0.14 | 0.41 | 0.47 | 67.4 | 0.28 | 0.38 | 0.43 | 63.6 | 0.22 |
| Potato | 0.14 | 0.17 | 56.3 | 0.12 | 0.31 | 0.38 | 64.5 | 0.19 | 0.21 | 0.25 | 52.9 | 0.07 |
| Hot chips | 0.31 | 0.38 | 55.4 | 0.18 | 0.36 | 0.43 | 62.8 | 0.25 | 0.15 | 0.18 | 52.4 | 0.06 |
| Takeaway style foods | 0.08 | 0.10 | 51.5 | 0.05 | 0.21 | 0.26 | 61.0 | 0.15 | 0.24 | 0.29 | 57.8 | 0.16 |
| Sweet snack foods | 0.37 | 0.43 | 61.5 | 0.20 | 0.22 | 0.26 | 58.7 | 0.08 | 0.39 | 0.46 | 67.4 | 0.27 |
| Savoury snack foods | 0.32 | 0.37 | 58.0 | 0.18 | 0.31 | 0.37 | 61.6 | 0.20 | 0.28 | 0.32 | 62.6 | 0.18 |
| Crispbreads and crackers | 0.32 | 0.44 | 60.6 | 0.21 | 0.35 | 0.41 | 70.4 | 0.27 | 0.32 | 0.37 | 63.1 | 0.19 |
| Nuts and seeds | 0.31 | 0.37 | 60.2 | 0.20 | 0.48 | 0.56 | 65.7 | 0.26 | 0.33 | 0.39 | 62.0 | 0.20 |
| Butter and margarine | 0.47 | 0.55 | 69.3 | 0.32 | 0.39 | 0.45 | 68.0 | 0.30 | 0.43 | 0.49 | 65.2 | 0.28 |
| Sugars, jams and honey | 0.37 | 0.43 | 68.0 | 0.27 | 0.22 | 0.26 | 63.4 | 0.18 | 0.27 | 0.32 | 67.4 | 0.18 |

†Spearman's correlation $r_s$, De-attenuated Spearman's correlation $R_S$, (<0.2 poor, 0.2–0.49 acceptable, ≥0.5 good), Percentage agreement within 1 quintile (≥50% good), $K_w$: weighted kappa (<0.20 Poor, 0.20–0.59 acceptable, ≥0.60 good)

all three time points, the FFQ produced higher estimates of intake for the majority of nutrients and food groups that were examined compared with the 24-hour recalls data. The FFQ demonstrated good-acceptable validity for some nutrients (e.g. protein, fat, fibre, iron, vitamin C, folate) and food groups (e.g. water, eggs, potato, hot chips, yoghurt, etc) at group level. Moreover, good-acceptable validity at individual level was revealed for most nutrients and food groups.

Even though the purpose designed FFQ overestimated intake of most nutrients and food intakes, it exhibited good-acceptable outcome for correlation and categorical agreement at individual level. Our findings are comparable with other studies that assessed both nutrient and food intakes in young children [37–42]. These studies have demonstrated that FFQs varied in estimating absolute intakes, but were able to rank participants adequately regardless of the age of the children, reference method used and length of the FFQ, a finding consistent with other population groups [43–45]. For example, Marriot et al. [40], evaluated a 78-item FFQ against 4 day weighed diaries in 12 month old infants demonstrating reasonable agreement (correlation: r = 0.25 to 0.66) relating to ranking of intakes despite differences in absolute

intake. In a Norwegian study, a semi-quantitative FFQ was validated against 7-day weighed food record among 12 month old children (n = 64), reporting higher energy intake and all nutrients except calcium, and variable median correlations between nutrient (0.18–0.72, median 0.50) and food intakes (0.28–0.83, median 0.62 with 38% of infants were classified in the same quartile [46]. Likewise, Buch-Andersen et al. [38] assessed the validity of a semi-quantitative FFQ among 3–9 year old Danish children (n = 54) also found that FFQ generally overestimated intakes relative to food records, and the correlations between two methods ranged from 0.29 to 0.63 for food intakes and 0.12 to 0.48 for nutrient intakes. Similarly, a short 10-item FFQ designed to assess fruit, vegetables, sugary foods and beverages in low-income children aged 2–4 years (n = 70), when compared to three non-consecutive 24-hour recalls, did not perform well regarding absolute food intakes, but it showed medium to strong validity in ranking participants with respect to food intakes (correlation: r = 0.30 to 0.59) [42].

Dietary assessment tools that have been designed specifically to measure dietary intake of young children in Australia are limited, and have often focused on development of short tools or on assessing specific foods groups rather than providing an assessment of total diet including food and nutrient intake. Flood et al. [17], developed a 17-item FFQ for 2–5 year old children (n = 77) to assess intakes of fruits, vegetables, lean and processed meats, take-away food, beverages and discretionary foods with evaluation against 3-day food records. That FFQ demonstrated moderate validity, with correlations ranging from 0.14 to 0.67 and >0.5 for vegetables, fruit, diet soft drinks and fruit juice [17]. Similarly, evaluation of a 19-item parent FFQ to measure diet quality among 12–36 month old children (n = 111) against a validated 54-item FFQ from Belgium (modified for Australia), revealed comparable diet quality scores [19]. In an earlier study, Bennett and colleagues evaluated a questionnaire focusing on 10 obesity-related food and beverage items with one single 24-hour recall among 2–5 year old Australian children (n = 90) and reported an acceptable level of relative validity (correlation: r = 0.57–0.88) [18]. Only Collins et al. [20], has developed and evaluated a comprehensive, 120 item FFQ for use in Australian toddlers which was evaluated using doubly-labelled water and demonstrated good agreement at the group level.

In contrast to the existing FFQ validation studies among young children that mostly utilised correlation coefficient and categorical agreement as primary indicators of validity, our study employed other validation tests to assess both group-level and individual-level agreement[34]. Three group-level and three individual-level validation tests provide further insight on the utility of the FFQ for examining dietary intake for groups (e.g. in research or public policy work) or individuals (e.g. in a clinical setting) [35]. The findings that our tool showed good-acceptable validity for some nutrients and food groups at group-level are expected as reference portion size from national surveys was used. Using reported portion size may improve the agreement of the FFQ and the 24 hour recalls measures in the present cohort. Nevertheless, good-acceptable validity at individual level for most nutrients and food groups was also revealed, highlighting the robustness of our tool for assessing individual dietary intake in clinical settings and ranking individuals according to nutrient and food intakes. Validation of this tool in a representative Australian sample may yield greater validity at both group- and individual-level. Although our FFQ validation is nested in an early childhood obesity prevention study, the children involved in our study were not overweight or obese at the start of the trial. We made no exclusion of children in terms of their birth weight or nutritional status. Therefore, the utility of our FFQ is not limited to programs seeking to prevent overweight and obesity only and it has validity in assessing dietary intake among general Australian children. When utilisation this FFQ tool in future studies researchers should be aware that the FFQ aims to rank subjects rather than provide true estimates of usual intakes. Special consideration on

measurement error and dietary misreporting for analysing and interpreting self-reported data including FFQ in dietary surveillance and nutritional epidemiology is needed [47].

Strengths of this study include the use of national survey data to inform development of the FFQ, and inclusion of objective measures of intake [11], a relatively large population-based sample and validation at multiple ages. Development of this FFQ was informed by national dietary data of Australian children and a list of commonly consumed foods was identified to capture >80% of total diet. This FFQ was validated against the use of three non-consecutive days of 24-hour recall covering both weekdays and weekend days. [11]. Further, the current study included a relatively large sample size in comparison to other validation studies among young children and the tool was validated across three different time points using a range of statistical techniques as recommended in the literature.

However, it is important to consider the limitations with this study, including the potential lack of generalisability with respect to the study sample, and the lack of inclusion of portion sizes in the FFQ. While the study sample was recruited using a population-based sampling frame, the participants included a relatively high proportion of mothers with a university education, increasing over subsequent waves of follow-up. While validity may improve due to higher literacy, more highly educated parents may also be subject to greater social desirability bias [48, 49]. Future studies should be conducted in populations of different sociodemographic background. To ease subjective burden and ensure generalisability of the FFQ at the population level, we did not ask respondents to estimate portion size in the FFQ and used reference portion sizes from a national survey to estimate intakes. This may have introduced significant error and resulted in the higher absolute intake of nutrient and foods from the FFQ. However, higher intakes estimated by FFQs is a consistent finding across other studies [44], suggesting this is not the sole reason for higher intakes. Of note, foods that revealed similar measures between two methods were mostly foods easily reported in specific units and have less variable portion size (e.g. bread). In addition, 24-hour recalls have less ability to account for day-to-day variation in food intake and capture less frequently consumed foods. Indeed, less frequently consumed foods, such as takeaway foods had a lower correlation between two methods. Additional days of reporting may have been beneficial, however the impact on subject burden and response rates is a major consideration, particularly in longitudinal studies, and with participants with major time constraints such as parents with young children [14]. Another limitation of the study is the gap between measurement of 24-hour recalls and FFQ. Although the number of days between two measures were less than a week, it could potentially affect the variation in intake and results in difference between two instruments. Pre-scheduled interview plus food diary was used to capture dietary intake from children caring by more than one carer (4%). However, the impact of this alternative approach on the results is likely to be small given the small percentage. Other limitations include those common to many FFQs, such as difficulties in the estimation of foods in mixed dishes that are difficult to quantify and may not be well captured in FFQs [50].

## Conclusion

This study evaluated a quantitative FFQ to assess the nutrient and food group intakes in young children aged 1.5 to 5 years. At all three time points, and consistent with other studies, the FFQ produced higher estimates of intake for the majority of nutrients and food groups that were examined when compared with estimates derived from 24-hour recalls. However, good-acceptable validity was demonstrated for some nutrients and food groups at group level, and most nutrients and food groups at individual level. The quantitative FFQ is suitable for monitoring dietary intake among young children under the five years of age in both research and

clinical settings. The utility of this FFQ to assess dietary intake at individual level to rank individuals according to nutrient (both macronutrient and key nutrients) and food intakes is however more robust than at group level. However, further testing in diverse population groups is warranted and assessment of portion sizes may further improve the validity of the tool.

## Supporting information

**S1 Fig. Assessment period between FFQ and three 24 hour recalls in the InFANT study.**
(DOCX)

**S1 Table. Comparison of children included and excluded from the analysis.**
(DOCX)

**S2 Table. Categorisation of 26 food groups used in comparison of FFQ with the 24-hour recalls.**
(DOCX)

**S3 Table. Comparison of energy-adjusted food intakes as estimated by the FFQ and the 24-hour recalls.**
(DOCX)

**S1 File. InFANT FFQ questionnaire.**
(PDF)

## Author Contributions

**Conceptualization:** Karen J. Campbell, Sarah A. McNaughton.

**Formal analysis:** Miaobing Zheng, Emily Scanlan, Sarah A. McNaughton.

**Funding acquisition:** Karen J. Campbell, Sarah A. McNaughton.

**Investigation:** Miaobing Zheng, Emily Scanlan, Sarah A. McNaughton.

**Methodology:** Miaobing Zheng, Karen J. Campbell, Emily Scanlan, Sarah A. McNaughton.

**Project administration:** Karen J. Campbell, Emily Scanlan, Sarah A. McNaughton.

**Resources:** Sarah A. McNaughton.

**Supervision:** Karen J. Campbell, Sarah A. McNaughton.

**Writing – original draft:** Miaobing Zheng, Sarah A. McNaughton.

**Writing – review & editing:** Miaobing Zheng, Karen J. Campbell, Emily Scanlan, Sarah A. McNaughton.

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
