## [Decision Letter · Decision Letter 0]

13 Nov 2019

PONE-D-19-24716

Development and evaluation of a food frequency questionnaire for use among young children

PLOS ONE

Dear Dr Zheng,

Thank you for submitting your manuscript to PLOS ONE. After careful consideration, we feel that it has merit but does not fully meet PLOS ONE’s publication criteria as it currently stands. Therefore, we invite you to submit a revised version of the manuscript that addresses the points raised during the review process.

Specifically, major concerns are pointed for methodological description flaws present in the manuscript. I would like to stress the points that reviewer #2 has found. More specifically, the context of your results in light of already available similar studies have been pointed out, and need to be solved

We would appreciate receiving your revised manuscript by Dec 28 2019 11:59PM. To enhance the reproducibility of your results, we recommend that if applicable you deposit your laboratory protocols in protocols.io, where a protocol can be assigned its own identifier (DOI) such that it can be cited independently in the future. For instructions see: http://journals.plos.org/plosone/s/submission-guidelines#loc-laboratory-protocols

We look forward to receiving your revised manuscript.

Kind regards,

Jose M. Moran

Academic Editor

PLOS ONE

Journal Requirements:

Reviewers' comments:

Reviewer's Responses to Questions

**Comments to the Author**

1. Is the manuscript technically sound, and do the data support the conclusions?

Reviewer #1: Yes

Reviewer #2: Partly

2. Has the statistical analysis been performed appropriately and rigorously? 

Reviewer #1: Yes

Reviewer #2: No

3. Have the authors made all data underlying the findings in their manuscript fully available?

Reviewer #1: No

Reviewer #2: Yes

4. Is the manuscript presented in an intelligible fashion and written in standard English?

Reviewer #1: Yes

Reviewer #2: Yes

5. Review Comments to the Author

Reviewer #1: This is a well-written manuscript. The analyses are well thought out and are in line with the literature to validate a FFQ.

The conclusion and the discussion reflect the proper future usage of the tool: to assess and rank dietary intake within a given population.

I can see that the response to previous reviewers was thorough and improved the manuscript.

My only comment is that the title should reflect the relative assessment of the tool. Ex. Development and relative evaluation of a FFQ for use among young children.

Also, the term 'validation' could replace 'evaluation' in the title.

Reviewer #2: This is a well-written manuscript detailing the development and validation of a food frequency questionnaire (FFQ) for use in young children, with a focus on foods associated with obesity, comparing the performance of the FFQ to 24-hr dietary recalls. Overall, the study procedures are rigorous. While most FFQs do not yield estimates of usual daily intake for nutrients or food groups. However, the authors have developed a quantitative QFFQ, a method that is becoming increasingly used. However, as noted below, there are further validation analyses and interpretation approaches for QFFQs that would greatly strengthen this manuscript and likely provide more information about the best uses of this FFQ.

INTRODUCTION:

The authors list other FFQs that have been developed, and it is unclear why these don’t suffice – why did the author feel the need to develop another FFQ?

METHODS:

The authors have developed at quantitative FFQ, which unlike non-quantitative FFQs, yields estimates of usual daily intake for nutrients and food groups. There are several group- and individual-level validation tests that can be employed to evaluate how valid the QFFQ results are for examination of individuals (e.g., in a clinical setting) or groups (e.g., in research or public policy work). Although the authors applied some of these methods, applying all of them and interpreting the results accordingly would greatly strengthen this manuscript. An articles that employs and describes these methods and their interpretation is:

Carter RC, Jacobson SW, Booley S, Najaar B, Dodge NC, Bechard LJ, Meintjes EM,Molteno CD, Duggan CP, Jacobson JL, Senekal M. Development and validation of a quantitative choline food frequency questionnaire for use with drinking and non-drinking pregnant women in Cape Town, South Africa. Nutr J. 2018 Nov 22;17(1):108. doi: 10.1186/s12937-018-0411-5. PubMed PMID: 30466439.

In addition, it is unclear why FFQ values were adjusted for energy or how this was done; was the energy from the 24-hr recall used to adjust FFQ values? How did the FFQ yield energy intake?

Why was the median portion size used instead of asking about a normal portion for that subject? This may have added a large degree of noise.

Why were Spearman correlations used only for the food group analyses and not the analysis of individual nutrients. Since non-quantitative FFQs are designed to compare subjects within a given cohort, Spearman correlations would demonstrate how this QFFQ could be used as a regular FFQ.

It is my impression that the formulas for de-attenuating Spearman and Pearson coefficients are different – see Willet’s textbook Nutritional Epidemiology.

DISCUSSION:

I would temper the validity and practical of this FFQ given that most of the tests uses yielded “fair” results. Perhaps the use of more validation tools will allow for more detailed interpretation (e.g., individual- vs group-level use) and bolster the discussion.

More detail should be given regarding the utility of this FFQ for use in obesity-targeting programs. It may be less valid in research or clinical programs with other aims (e.g., undernutrition).

6. PLOS authors have the option to publish the peer review history of their article (what does this mean?). If published, this will include your full peer review and any attached files.

Reviewer #1: Yes: Valerie Marcil

Reviewer #2: No

---

## [Author Response · Author response to Decision Letter 0]

19 Dec 2019

Note: all page numbers refer to the manuscript file with track changes

Reviewer #1: This is a well-written manuscript. The analyses are well thought out and are in line with the literature to validate a FFQ.

The conclusion and the discussion reflect the proper future usage of the tool: to assess and rank dietary intake within a given population.

I can see that the response to previous reviewers was thorough and improved the manuscript.

My only comment is that the title should reflect the relative assessment of the tool. Ex. Development and relative evaluation of a FFQ for use among young children.

Also, the term 'validation' could replace 'evaluation' in the title.

Reply: Thank you for your time in reviewing our paper and valuable feedbacks. We did carefully consider the use of wording ‘evaluation’ over ‘validation’. Validation of dietary intake often refers to comparing against true intakes (an unbiased reference). As the reference method: three-24hr recalls utilised in our study are self-report measures that capture intake with some degree of error or inaccuracy, we therefore, used ‘evaluation’ over ‘validation’. The wording ‘validated FFQ’ should be used with caution despite that this has been widely misused in the literature. We are aware some studies make this distinction by using ‘relative validation’. However, we think it’s important to keep this distinction more apparent by using ‘evaluation’. Thanks again for your suggestion.

Please refer to the following to papers regarding the use of “ a validated FFQ”.

Frongillo et al. Establishing Validity and Cross-Context Equivalence of Measures and Indicators J Acad Nutr Diet. 2019 Nov;119(11):1817-1830. https://www.ncbi.nlm.nih.gov/pubmed/30470590

Kirkpatrick et al. Best Practices for Conducting and Interpreting Studies to Validate Self-Report Dietary Assessment Methods. J Acad Nutr Diet. 2019 Nov;119(11):1801-1816. https://www.ncbi.nlm.nih.gov/pubmed/31521583

Reviewer #2: This is a well-written manuscript detailing the development and validation of a food frequency questionnaire (FFQ) for use in young children, with a focus on foods associated with obesity, comparing the performance of the FFQ to 24-hr dietary recalls. Overall, the study procedures are rigorous. While most FFQs do not yield estimates of usual daily intake for nutrients or food groups. However, the authors have developed a quantitative QFFQ, a method that is becoming increasingly used. However, as noted below, there are further validation analyses and interpretation approaches for QFFQs that would greatly strengthen this manuscript and likely provide more information about the best uses of this FFQ.

Reply: Thank you for your time in reviewing our paper and valuable feedbacks.

INTRODUCTION:

 The authors list other FFQs that have been developed, and it is unclear why these don’t suffice – why did the author feel the need to develop another FFQ?

Reply: Thank you for raising this important point. The introduction has now been amended to reflect the need to develop a new tool and motivate our aim.

Several FFQs have been developed for young Australian children to assess intakes of specific food groups or nutrients rather than total diet including both absolute food and nutrient intakes [17-21]. Flood et al., [17] and Bennett et al., [18] developed and evaluated short FFQs for measuring daily serving of food and beverage intakes. Devenish et al., [21] validated a tool to assess total and free sugars intakes of Australian toddlers. Only one quantitative FFQ exists [20], but the tool was not developed specifically for young children, rather it was modified from a quantitative FFQ for older children and adolescents. Therefore, a quantitative FFQ developed specifically for Australian young children to assess daily intake of both nutrients and food groups is needed. (line 66-76)

Detailed comparison of our tool to the existing Australian FFQ is also provided in the discussion (line 423-437).

 The authors have developed at quantitative FFQ, which unlike non-quantitative FFQs, yields estimates of usual daily intake for nutrients and food groups. There are several group- and individual-level validation tests that can be employed to evaluate how valid the QFFQ results are for examination of individuals (e.g., in a clinical setting) or groups (e.g., in research or public policy work). Although the authors applied some of these methods, applying all of them and interpreting the results accordingly would greatly strengthen this manuscript. An articles that employs and describes these methods and their interpretation is:

Carter RC, Jacobson SW, Booley S, Najaar B, Dodge NC, Bechard LJ, Meintjes EM,Molteno CD, Duggan CP, Jacobson JL, Senekal M. Development and validation of a quantitative choline food frequency questionnaire for use with drinking and non-drinking pregnant women in Cape Town, South Africa. Nutr J. 2018 Nov 22;17(1):108. doi: 10.1186/s12937-018-0411-5. PubMed PMID: 30466439.

Reply: Thank you for your comments on group- and individual-level validation tests and providing reference to guide us. We have now conducted two additional group-level validation tests (Bland-Altman correlation coefficient between mean and mean difference and limit of agreement index) to strengthen the validity of our tool and added Carter et al and Lombard et al as references. We also followed the interpretation suggested by Carter et al and Lombard et al. Tables have been revised to present results of group level and individual level tests, respectively. All sections of the paper have now been revised accordingly to present validation analyses and interpretation by group-level and individual-level. Please see the following edits in the paper:

Lombard, M.J.; Steyn, N.P.; Charlton, K.E.; Senekal, M. Application and interpretation of multiple statistical tests to evaluate validity of dietary intake assessment methods. Nutrition journal 2015, 14, 40, doi: 10.1186/s12937-015-0027-y. https://www.ncbi.nlm.nih.gov/pmc/articles/PMC4471918/

Abstract:

Results: Out of 11 nutrients, four (protein, fat, fibre, iron), two (Vitamin C, folate) and three (protein, vitamin C and folate) nutrients showed good-acceptable outcome for 2 out of 3 group-level validation tests at ages 1.5, 3.5 and 5.0 years, respectively. Of 26 food groups, good-acceptable outcome for 2 out of 3 group-level validation tests was revealed for two, four and six food groups at ages 1.5, 3.5 and 5.0 years, respectively. For individual-level validation tests, all nutrients showed good-acceptable outcome for 2 out of 3 individual level tests across three time points, except for folate at age 1.5 years and energy intake at age 3.5 years. Most food groups (22 out of 26) at age 1.5 years and all food groups at both ages 3.5 and 5.0 years showed good-acceptable outcome for 2 out of 3 individual-level validation tests. (line 30-38)

Conclusion: At all three time points, the FFQ demonstrated good-acceptable validity for some nutrients and food groups at group-level, and good-acceptable validity for most nutrients and food groups at individual-level. This quantitative FFQ is a valid and robust tool for assessing total diet of young children and ranking individuals according to nutrient and food intakes. (line 42-45)

Methods:

Group-level and individual-level validation tests with implications for research and clinical setting, respectively, were conducted to test differences between the FFQ and 24-hour recall measures [Cater et al, Lombard et al]. Three group-level validation tests were performed: paired t-test/Wilcoxon signed rank sum test, Bland-Altman correlation between mean and mean difference, and Bland-Altman limit of agreement (LOA) index. (line 177-181).

Bland-Altman correlation between mean and mean difference of two measures examines the presence of proportional bias as well as agreement at the group level with a significant P-value ≤0.05 indicating the presence of bias. Bland-Altman LOA index was calculated as the percentage of individuals with values outside of the limit of agreement (<5% being acceptable) [34]. The individual-level agreement was assessed by three validation tests: correlation coefficient, categorical agreement and weighted kappa. Pearson’s and Spearman’s correlation (r) analyses were performed for nutrient and food group intakes respectively to assess strength and direction of association at individual level. Strength of correlation was defined as poor (<0.20), acceptable (0.20-0.49), and good (≥0.50) outcomes. (line 184-193).

Results:

For Bland-Altman correlation of mean and mean difference, most nutrients at ages 1.5 (8 out of 11) at 5 (6 out of 11) years exhibited no proportional bias (P>0.05). In contrast, at age 3.5 years, 9 out of 11 nutrients revealed bias. Of 11 nutrients assessed, 6, 5 and 4 nutrients showed acceptable outcome for Bland-Altman LOA index at age 1.5, 3.5 and 5 years, respectively. Four (protein, fat, fibre, iron), two (Vitamin C, folate) and three (protein, vitamin C and folate) nutrients at ages 1.5, 3.5 and 5.0 years, respectively, showed good-acceptable outcome for 2 out of 3 group-level validation tests. (line 228-234).

Table 3 illustrates the individual-level validation test results for FFQ and 24-hour recall energy-adjusted nutrient intakes. Out of 11 nutrients assessed, most nutrients exhibited acceptable correlation (r>0.20) at ages 1.5 (n=10), 3.5 (n=9) and 5.0 (n=10) years. The mean Pearson’s r for nutrient intakes was 0.30, 0.33 and 0.32 for ages 1.5, 3.5 and 5.0 years, respectively. Comparison of equality of average Pearson’s r between time points revealed no significant difference. Accounting for within person variation, the Pearson correlation at all time points for all nutrients improved, and the average de-attenuated Pearson’s R was 0.34, 0.38 and 0.37 at ages 1.5, 3.5 and 5 years, respectively. Percentage agreement of nutrient intakes between the FFQ and recall (agreement within 1 quintile) showed good agreement (>50%) with average percentage agreement of 62.7%, 64.7%, and 62.4%, at age 1.5, 3.5, 5.0 years, respectively. Weighted kappa values showed agreement ranging from poor to acceptable with Kw of 0.06 to 0.34 across three time point. The mean Kw at age 1.5, 3.5 and 5.0 years was 0.19, 0.21, and 0.19, respectively. Ten out of 11 nutrients at ages 1.5 and 3.5 years, and all nutrients at age 5.0 years exhibited good-acceptable agreement for 2 out of 3 individual-level validation tests. The number of nutrients showed good-acceptable agreement for all three individual level tests was 3, 7, and 5 at ages 1.5, 3.5 and 5.0 years, respectively. (line 235-247)

Results of two other group-level validation tests are reported in Table 4. The number of food groups showed no proportional bias (Bland-Altman correlation P≤0.05) was fairly consistent across time point (9, 10 and 9 out of 26 respectively for ages 1.5, 3.5 and 5.0 years). For Bland-Altman LOA index, 7, 9, 12 food groups at ages 1.5, 3.5 and 5.0 years, respectively showed acceptable outcome (< 5%). Two (water, sugars/jams/honey), four (eggs, potato, hot chips, crispbread/crackers) and six (all other beverage, yoghurt, cream/ice cream/custard, white bread, poultry, takeaway style foods) food groups at ages 1.5, 3.5 and 5.0 years, respectively, showed good-acceptable outcome for 2 of 3 group-level validation tests. (line 304-311).

Individual-level validation analyses results for energy-adjusted food intakes estimated by FFQ and 24-hour recall are presented in Table 4. Mean Spearman’s r among all food groups was 0.31, 0.34, and 0.34 respectively at ages 1.5, 3.5 and 5.0 years. All food groups at age 3.5 years and most food groups at ages 1.5 (21/26) and 5.0 (24/26) years revealed acceptable correlations (r≥0.20). No significant difference between Spearman’s r between time points were found. Spearman correlation for all food intakes at all time points improved after correcting for within person variation with average de-attenuated Spearman’s R of 0.36, 0.40, and 0.39 at ages 1.5, 3.5, and 5.0 years, respectively. The percentage agreement of all food intakes (agreement within 1 quintile) across all time points was above 50% with average percentage agreement of 62.2%, 64.0% and 64.2% at age 1.5, 3.5 and 5.0 respectively (Table 5). Weighted kappa for food intakes showed poor to acceptable agreement ranging from 0.05 to 0.47 across three time point. The mean Kw value at age 1.5, 3.5 and 5.0 years was 0.21, 0.22, and 0.22 respectively. Most food groups (22 out of 26) at age 1.5 years and all food groups at both ages 3.5 and 5.0 years showed good-acceptable outcome for 2 out of 3 individual-level validation tests. Similar number of food groups (15 at age 1.5 years, 14 at age 3.5 years and 13 at age 5.0 years) revealed good-acceptable outcome for all three individual-level validation tests across time points.(line 312-326)

Discussion:

The FFQ demonstrated good-acceptable validity for some nutrients (e.g. protein, fat, fibre, iron, vitamin C, folate) and food groups (e.g. water, eggs, potato, hot chips, yoghurt, etc) at group level. Moreover, good-acceptable validity at individual level was revealed for most nutrients and food groups. (line 396-399)

In contrast to the existing FFQ validitation studies among young children that mostly utilised correlation coefficient and categorical agreement as primary indicators of validity, our study employed other validation tests to assess both group-level and individual-level agreement(Lombard et al). Three group-level and three individual-level validation tests provide further insight on the utility of the FFQ for examining dietary intake for groups (e.g. in research or public policy work) or individuals (e.g. in a clinical setting) [34]. The findings that our tool showed good-acceptable validity for some nutrients and food groups at group-level are expected as reference portion size from national surveys was used. Using reported portion size may improve the agreement of the FFQ and the 24 hour recalls measures in the present cohort. Nevertheless, good-acceptable validity at individual level for most nutrients and food groups was also revealed, highlighting the robustness of our tool for assessing individual dietary intake in clinical settings and ranking individuals according to nutrient and food intakes. Validation of this tool in a representative Australian sample may yield greater validity at both group- and individual-level. (line 437-449)

Conclusion 

However, good-acceptable validity was demonstrated for some nutrients and food groups at group level, and most nutrients and food groups at individual level. The quantitative FFQ is suitable for monitoring dietary intake among young children under the five years of age in both research and clinical settings. The utility of this FFQ to assess dietary intake at individual level to rank individuals according to nutrient (i.e. macronutrient and key nutrients) and food intakes is however more robust than at group level. However, further testing in diverse population groups is warranted and assessment of portion sizes may further improve the validity of the tool. (line 493-500). 

 In addition, it is unclear why FFQ values were adjusted for energy or how this was done; was the energy from the 24-hr recall used to adjust FFQ values? How did the FFQ yield energy intake?

Reply: All self-report dietary assessment methods are subject to measurement error. The main purpose for adjusting total energy intake is to remove some of the measurement errors. It is recommended to use self-reported energy intake for energy adjustment of other self-reported dietary constituents to improve risk estimation in studies of diet-health associations and measurements of validity should also be adjusted for energy intake”. For example, correlation coefficient may be inflated due to correlated errors, by using energy-adjusted nutrient and food may reduce correlated errors.

Please refer to the following reference for further information:

Willett WC. Nutritional Epidemiology. New York: Oxford University Press; 2012.

Subar et al. Addressing Current Criticism Regarding the Value of Self-Report Dietary Data.J Nutr. 2015 Dec;145(12):2639-45. https://www.ncbi.nlm.nih.gov/pubmed/26468491

McNaughton et al. Validation of a food-frequency questionnaire assessment of carotenoid and vitamin E intake using weighed food records and plasma biomarkers: the method of triads model. Eur J Clin Nutr. 2005 Feb;59(2):211-8.https://www.ncbi.nlm.nih.gov/pubmed/15483635

The following edits now been added: Nutrient and food intakes from both FFQ and 24-hour recall were energy-adjusted to remove correlated measurement error using the residual method with nutrient and food intakes regressed on total energy intake [37]. (line 202=204). 

Please see the following lines for deriving energy intake from FFQ:

Food and nutrient intakes from the FFQ were calculated using a purposed designed database incorporating AUSNUT2007 food composition database (line 135-136). Daily consumption of each FFQ food item in grams was calculated by converting the frequency of consumption into daily equivalents (Never=0; 1-3/month=0.067; 1/week=0.143; 2-4/week=0.429; 5-6/week=0.786; 1/day=1.0; 2-3/day=2.5; 4-5/day=4.5; ≥6/day=6.0) and then multiplying by the calculated median portion size for that food (line 142-145).

 Why was the median portion size used instead of asking about a normal portion for that subject? This may have added a large degree of noise.

Reply: To ease subjective burden and ensure the generalisability of the FFQ at the population level, the FFQ did not ask respondents to report portion size. Instead, the reference portion size calculated from the Australian National Health survey was used. We acknowledge this may have led to the higher absolute food and nutrient intakes estimated by FFQ compared to the 24 hour recalls. This is discussed in the limitation section in line 468-472.

 Why were Spearman correlations used only for the food group analyses and not the analysis of individual nutrients. Since non-quantitative FFQs are designed to compare subjects within a given cohort, Spearman correlations would demonstrate how this QFFQ could be used as a regular FFQ.

Reply: Pearson correlation of individual nutrients was also conducted, and results are now presented in Table 3 (please see r and de-attenuated R column). Line 236-242 also described the Pearson correlation between FFQ and 24 hr recall for nutrient intakes. As most nutrient intakes were normally distributed and food intakes were skewed, Pearson and Spearman correlation was used for food and nutrients intakes, respectively.

 It is my impression that the formulas for de-attenuating Spearman and Pearson coefficients are different – see Willet’s textbook Nutritional Epidemiology.

Reply: We have now extracted the original formula for correction of correlation coefficient from the Walter willet book which is:

r_t=r_0 √(ʎ_x/n_x+ʎ_y/n_y ) , and ʎ_x and ʎ_y is the ratio of within- and between-person variances for x and y, respectively. Given that we are comparing one FFQ estimate (no within person variance) with three 24hour recall estimate, the formula reduced to : r_t=r_0 √(1+ʎ_x/n_x ) and n_x in our case is 3, resulting in our formula as R=r√(1+ʎ/3) (to avoid using letter with subscript r_t, we used R to represent de-attenuated correlation coefficient). We have modified this sentence, so the formula is more closely aligned with the Walter Willet book. Please see below:

To account for random within person variation among three 24-hour recalls, the energy-adjusted correlation coefficients were de-attenuated based on the formula:R=r√(1+ʎ_x/n_x ) in which the attenuation factor (ʎ_x ) is the ratio of within- and between-person variances obtained from the three recalls and n_x equals to 3 (line 193-196).

Willett WC. Nutritional Epidemiology. New York: Oxford University Press; 2012.

DISCUSSION:

 I would temper the validity and practical of this FFQ given that most of the tests uses yielded “fair” results. Perhaps the use of more validation tools will allow for more detailed interpretation (e.g., individual- vs group-level use) and bolster the discussion.

Reply: Thank you for your comments. Please see our response to comment 2 where we provided new interpretation.

 More detail should be given regarding the utility of this FFQ for use in obesity-targeting programs. It may be less valid in research or clinical programs with other aims (e.g., undernutrition).

Reply: Although our FFQ validation is nested in an early childhood obesity prevention study, the children involved in our study were not overweight or obese at the start of the trial. We made no exclusion of children in terms of their birth weight or nutritional status. Therefore, the utility of our FFQ is not limited to programs seeking to prevent overweight and obesity only and it has validity in assessing dietary intake among general Australian children. The following is added to the discussion: “The quantitative FFQ is suitable for monitoring dietary intake among young children under the five years of age in both research and clinical settings. The utility of this FFQ to assess dietary intake at individual level to rank individuals according to nutrient (both macronutrient and key nutrients) and food intakes is however more robust than at group level. However, further testing in diverse population groups is warranted and assessment of portion sizes may further improve the validity of the tool. “ (line 495-500)

---

## [Decision Letter · Decision Letter 1]

30 Jan 2020

PONE-D-19-24716R1

Development and evaluation of a food frequency questionnaire for use among young children

PLOS ONE

Dear Dr Zheng,

Thank you for submitting your manuscript to PLOS ONE. After careful consideration, we feel that it has merit but does not fully meet PLOS ONE’s publication criteria as it currently stands. Therefore, we invite you to submit a revised version of the manuscript that addresses the points raised during the review process.

Congratulations on the work done in reviewing your manuscript. Please address the reviewer's requirements as far as possible, particularly those that refer to the limitations of the study. 

We would appreciate receiving your revised manuscript by Mar 15 2020 11:59PM. To enhance the reproducibility of your results, we recommend that if applicable you deposit your laboratory protocols in protocols.io, where a protocol can be assigned its own identifier (DOI) such that it can be cited independently in the future. For instructions see: http://journals.plos.org/plosone/s/submission-guidelines#loc-laboratory-protocols

We look forward to receiving your revised manuscript.

Kind regards,

Jose M. Moran

Academic Editor

PLOS ONE

Reviewers' comments:

Reviewer's Responses to Questions

**Comments to the Author**

1. If the authors have adequately addressed your comments raised in a previous round of review and you feel that this manuscript is now acceptable for publication, you may indicate that here to bypass the “Comments to the Author” section, enter your conflict of interest statement in the “Confidential to Editor” section, and submit your "Accept" recommendation.

Reviewer #2: All comments have been addressed

2. Is the manuscript technically sound, and do the data support the conclusions?

Reviewer #2: Yes

3. Has the statistical analysis been performed appropriately and rigorously? 

Reviewer #2: Yes

4. Have the authors made all data underlying the findings in their manuscript fully available?

Reviewer #2: No

5. Is the manuscript presented in an intelligible fashion and written in standard English?

Reviewer #2: Yes

6. Review Comments to the Author

Reviewer #2: The authors have done a tremendous job at responding to the reviewers’ comments and requests, and the manuscript has been greatly strengthened. Only a few issues remain:

1) Per the Willett textbook quoted, Spearman correlation should be used for FFQ data, whether the data are normally distributed or not, as the FFQ aims to rank subjects within a cohort rather than to provide point estimates of usual intake.

2) Given the aim of FFQ to rank subjects rather than provide point estimates of usual intake, the energy intake calculations are at best gross estimates. The use of national median portion sizes also adds significant error to these estimates. These limitations should be more clearly emphasized in discussing that approach and in interpreting the results.

3) The authors’ response:

“Although our FFQ validation is nested in an early childhood obesity prevention study, the children involved in our study were not overweight or obese at the start of the trial. We made no exclusion of children in terms of their birth weight or nutritional status. Therefore, the utility of our FFQ is not limited to programs seeking to prevent overweight and obesity only and it has validity in assessing dietary intake among general Australian children….” Is very convincing and should be included in the Discussion, as many readers will have the same question.

7. PLOS authors have the option to publish the peer review history of their article (what does this mean?). If published, this will include your full peer review and any attached files.

Reviewer #2: No

---

## [Author Response · Author response to Decision Letter 1]

18 Feb 2020

1) Per the Willett textbook quoted, Spearman correlation should be used for FFQ data, whether the data are normally distributed or not, as the FFQ aims to rank subjects within a cohort rather than to provide point estimates of usual intake.

Reply: Thank you for your advice. We agree that FFQ aims to rank subjects within a cohort rather than to provide point estimate of usual intake, and that Spearman correlation can indeed be used. As outlined below, however, most studies used Pearson correlation to compare nutrient intakes, and therefore we used Pearson correlation for consistency and comparison with other studies. Moreover, generally for two normally distributed variables, the results from the Pearson and Spearman values will be quite similar when the relationship is linear (usually reasonably true in the case of validation studies (Reference: Willett WC. Nutritional Epidemiology. New York: Oxford University Press; 2012. Chapter 6. DOI 10.1093/acprof:oso/9780199754038.003.0006, page 68 out of 91)

We carefully considered the type of validation test in our study and followed the recommendation from the Willet textbook and Margetts textbook to use Pearson Correlation to compare nutrient intakes between FFQ and the reference method (i.e. 24hr recall measures in our study) when data are normally distributed – and to use Spearman correlation to compared skewed food intakes (rational provided below). In both Chapter 6 of Willet’s textbook and Chapter 8 of the Margetts’s textbook, all examples on validity of FFQ nutrient intake against measures from the reference methods utilized Pearson Correlation. 

The Margetts textbook notes: “The most common method of assessing the validity of questionnaires is to test the agreement in ranking of subjects between questionnaire and standard. Consistency of ranking is usually measured using the Pearson product−moment correlation coefficient (often on log transformed data to improve approximation to the normal distribution), although other correlation coefficients (Spearman’s, intra-class) are also used.” (Reference: Margetts BM, Nelson M. Design Concepts in Nutritional Epidemiology. Oxford University Press; 1997. DOI:10.1093/acprof:oso/9780192627391.003.0008, Chapter 8. (8.6.2). Ranking and regression, page 24 out of 43).

Further support for the use of Pearson correlation in FFQ validation studies has also been found in Lombard et al., who reviewed the range of statistical tests used in the validation of quantitative FFQ and indicated that Pearson Correlation is one of the main tests used in FFQ validation studies to assess the correlation coefficient. (References: Lombard MJ, Steyn NP, et al. Application and interpretation of multiple statistical tests to evaluate validity of dietary intake assessment methods. Nutrition Journal. 2015: 14, 40)

In considering skewed data, Willett notes: “Because dietary variables are usually skewed toward higher values, transformations (such as log) to increase normality should be considered before computing correlation coefficients. This has the advantage of reducing the influence of extreme values and of creating a correlation coefficient that can be interpreted in the form of a contingency table. Alternatively, nonparametric correlation coefficients (e.g., Spearman) can be used when one or both variables are not normally distributed.” We provide this quote to justify our use of spearman correlation to compare skewed food intakes. (Reference: Willett WC. Nutritional Epidemiology. New York: Oxford University Press; 2012. Chapter 6. DOI 10.1093/acprof:oso/9780199754038.003.0006, page 71 out of 91).

2) Given the aim of FFQ to rank subjects rather than provide point estimates of usual intake, the energy intake calculations are at best gross estimates. The use of national median portion sizes also adds significant error to these estimates. These limitations should be more clearly emphasized in discussing that approach and in interpreting the results.

Reply: Thank you for your comments. Please see edits below to further highlight the limitations and assist readers with use of our tool in future studies. 

When utilization this FFQ tool in future studies researchers should be aware that the FFQ aims to rank subjects rather than provide true estimates of usual intakes. Special consideration on measurement error and dietary misreporting for analysing and interpreting self-reported data including FFQ in dietary surveillance and nutritional epidemiology is needed (Subar et al). Now added as Line 362-365 

To ease subject burden and ensure generalisability of the FFQ at the population level, we did not ask respondents to estimate portion size in the FFQ and used reference portion sizes from a national survey to estimate intakes. This may have introduced significant error and resulted in the higher absolute intake of nutrient and foods from the FFQ. (in Line 383-387)

Subar A. et al. Addressing Current Criticism Regarding the Value of Self-Report Dietary Data.J Nutr. 2015 Dec;145(12):2639-45. https://www.ncbi.nlm.nih.gov/pubmed/26468491

3) The authors’ response:

“Although our FFQ validation is nested in an early childhood obesity prevention study, the children involved in our study were not overweight or obese at the start of the trial. We made no exclusion of children in terms of their birth weight or nutritional status. Therefore, the utility of our FFQ is not limited to programs seeking to prevent overweight and obesity only and it has validity in assessing dietary intake among general Australian children….” Is very convincing and should be included in the Discussion, as many readers will have the same question.

Reply: Thank you for your suggestion. These sentences have now been added to the discussion in line 357-361.

---

## [Editor Report · Decision Letter 2]

6 Mar 2020

Development and evaluation of a food frequency questionnaire for use among young children

PONE-D-19-24716R2

Dear Dr. Zheng,

We are pleased to inform you that your manuscript has been judged scientifically suitable for publication and will be formally accepted for publication once it complies with all outstanding technical requirements.

With kind regards,

Jose M. Moran

Academic Editor

PLOS ONE
---

## [Editor Report · Acceptance letter]

10 Mar 2020

PONE-D-19-24716R2 

Development and evaluation of a food frequency questionnaire for use among young children 

Dear Dr. Zheng:

I am pleased to inform you that your manuscript has been deemed suitable for publication in PLOS ONE. Congratulations! Your manuscript is now with our production department. 

With kind regards,

on behalf of

Dr. Jose M. Moran 

Academic Editor

PLOS ONE